# Thermoelectric Properties of Cu_2_Se Nano-Thin Film by Magnetron Sputtering

**DOI:** 10.3390/ma14082075

**Published:** 2021-04-20

**Authors:** Liangliang Yang, Jiangtao Wei, Yuanhao Qin, Lei Wei, Peishuai Song, Mingliang Zhang, Fuhua Yang, Xiaodong Wang

**Affiliations:** 1Engineering Research Center for Semiconductor Integrated Technology, Institute of Semiconductors, Chinese Academy of Sciences, Beijing 100083, China; yangliangliang@semi.ac.cn (L.Y.); weijt@semi.ac.cn (J.W.); weilei@semi.ac.cn (L.W.); pssong@semi.ac.cn (P.S.); zhangml@semi.ac.cn (M.Z.); fhyang@semi.ac.cn (F.Y.); 2College of Materials Science and Opto-Electronic Technology, University of Chinese Academy of Sciences, Beijing 100049, China; qinyuanhao17@mails.ucas.ac.cn; 3School of Electronic, Electrical and Communication Engineering, University of Chinese Academy of Sciences, Beijing 100190, China; 4The School of Microelectronics & Center of Materials Science and Optoelectronics Engineering, University of Chinese Academy of Sciences, Beijing 100049, China; 5Beijing Engineering Research Center of Semiconductor Micro-Nano Integrated Technology, Beijing 100083, China; 6Beijing Academy of Quantum Information Science, Beijing 100193, China

**Keywords:** Cu_2_Se, thermoelectrics, film, magnetron sputtering, *ZT*

## Abstract

Thermoelectric technology can achieve mutual conversion between thermoelectricity and has the unique advantages of quiet operation, zero emissions and long life, all of which can help overcome the energy crisis. However, the large-scale application of thermoelectric technology is limited by its lower thermoelectric performance factor (*ZT*). The thermoelectric performance factor is a function of the Seebeck coefficient, electrical conductivity, thermal conductivity and absolute temperature. Since these parameters are interdependent, increasing the *ZT* value has always been a challenge. Here, we report the growth of Cu_2_Se thin films with a thickness of around 100 nm by magnetron sputtering. XRD and TEM analysis shows that the film is low-temperature α-Cu_2_Se, XPS analysis shows that about 10% of the film’s surface is oxidized, and the ratio of copper to selenium is 2.26:1. In the range of 300–400 K, the maximum conductivity of the film is 4.55 × 10^5^ S m^−1^, which is the maximum value reached by the current Cu_2_Se film. The corresponding Seebeck coefficient is between 15 and 30 µV K^−1^, and the maximum *ZT* value is 0.073. This work systematically studies the characterization of thin films and the measurement of thermoelectric properties and lays the foundation for further research on nano-thin-film thermoelectrics.

## 1. Introduction

The global energy crisis and environmental problems caused by the burning of fossil fuels have aroused widespread concern about alternative energy sources. Thermoelectric (TE) materials can directly convert industrial waste heat into useful electrical energy only through solid-state methods. In addition, they can also be used as heat pumps to provide local cooling. Although they have many advantages, such as lightweight, reliability, complete solid-state, and long service life, the use of TE materials is limited mainly due to low conversion efficiency. So far, extensive research has been conducted to improve the thermoelectric efficiency, which is evaluated by the dimensionless quality factor *ZT* (*ZT* = *S*^2^*σT*/*κ*), where *σ* is the conductivity, *S* is the Seebeck coefficient, *T* is the absolute temperature, and *κ* is the total thermal conductivity, including the contribution of its electronic (*κ_e_*) and lattice (*κ_L_*) components. For ideal thermoelectric materials, high-power factor (*S*^2^*σ*) and low *κ* are required to obtain high *ZT*, ensuring high thermoelectric efficiency. However, due to the interdependence and conflict of Seebeck coefficient, electrical conductivity and thermal conductivity, how to optimize the various parameters of *S*, *σ* and *κ* is always a challenge. People have been working to improve *ZT* by adjusting the transmission of electrons and phonons through nanostructures. Nanostructures can reduce the thermal conductivity of phonons through interface scattering while simultaneously confinement or maintaining or even improving the transmission of electrons [1,2,3,4,5,6]. Indeed, in thin-film structures (such as superlattices) and nanostructures (nano bulk) materials, *ZT* in nanostructures has been significantly improved. Traditional thermoelectric materials are mainly based on Bi, Te, Sb, Pb and other elements, including typical Bi_2_Te_3_ inorganic materials, PbTe, SiGe, skutterudite and semi-Hosler alloys [7,8,9]. Although the *ZT* growth of Bi_2_Te_3_ has made significant progress in the past few decades, its relatively high cost and scarcity have prompted developing alternatives to Bi_2_Te_3_. In recent years, Cu_2_Se and its derivatives have become the subject of intense interest in developing thermoelectric materials with high *ZT* [10,11,12,13,14,15].

In 2017, Zhao et al. reported that through a combination of the lowered electrical and lattice thermal conductivities and the relatively good carrier mobility caused by the weak alloy scattering potential, ultrahigh *ZT* values are achieved in slightly S-doped Cu_2_Se with a maximal value of 2.0 at 1000 K, 30% higher than that in nominally stoichiometric Cu_2_Se [15]. In 2017, Nunna et al. reported that by using the special interaction between metallic copper and multi-walled carbon nanotubes, they successfully achieved in-situ growth of Cu_2_Se on the surface of carbon nanotubes and then prepared a series of highly Cu_2_Se/CNTs hybrid materials. The number of molecular CNTs uniformly dispersed in the Cu_2_Se matrix. Exceeding the traditional mixing rules, a substantial reduction in lattice thermal conductivity and carrier concentration was observed, resulting in a record-setting thermoelectric quality factor *ZT* of 2.4 at 1000 K. This work opens up a new window for optimizing thermoelectric properties through organic/inorganic hybridization of nano or molecular-level materials [16]. In 2017, Pammi et al. reported on the manufacture of Cu_2-x_Se NW-polyvinylidene fluoride (PVDF) composite flexible film, which uses a variety of methods, such as simple drop-casting (DC), vacuum filtration (VF) and vacuum filtration, then mechanical pressurization (VFMP). When mechanically pressurized after vacuum filtration at 303 K, the highest power factor of Cu_2-x_Se NW-PVDF composite membrane reached 105.32 mW m^−1^ K^−2^ and further increased to 253.49 mW m^−1^ K^−2^ at 393 K [17]. In 2018, Li et al. reported that a thermoelectric material composed of Cu_2_Se incorporated up to 0.45 wt% of graphene nanoplates. Carbon-enhanced Cu_2_Se has an ultra-high thermoelectric quality factor of *ZT* = 2.44 ± 0.25 at 870 K [18]. In 2019, Byeon et al. found that in metallic Cu_2_Se, near 350 K, larger PF values and *ZT* exceeding 2.3 W m^−1^ K^−2^ and 470, respectively, and the abnormal temperature dependence of Seebeck coefficient can be observed. DFT band calculations show that this huge Seebeck effect is caused by the self-tuning carrier concentration effect and the coexistence of the two phases during the phase transition [19]. In 2020, Yang et al. can stabilize Cu_2_Se by adjusting the behavior of Cu^+^ ions and electrons in the Schottky heterojunction between the Cu_2_Se host matrix and the BiCuSeO nanoparticles formed in situ. It is helpful to obtain an excellent *ZT* value at 973 K with a peak *ZT* value of about 2.7 and a high average *ZT* value of 1.5 between 400 and 973 K. This discovery provides a new way for stable hybrid ion–electron conduction thermoelectrics and provides new insights for controlling ion migration in these materials based on ion migration [11].

Although high-performance is achieved in a bulk form, there are few studies on the thermoelectric properties of Cu_2_Se thin films, and the measurement of the properties of the thin films has always been a challenge. In 2017, Lin et al. reported a flexible thermoelectric copper selenide film composed of earth-rich elements. Using ink solutions with truly soluble precursors, thin films are manufactured through a low-cost and scalable spin-coating process. The power factor of the Cu_2_Se film on the rigid Al_2_O_3_ substrate at 684 K is 0.62 mW m^−1^ K^−2^, while the power factor at 664 K on the flexible polyimide substrate is 0.46 mW m^−1^ K^−2^. The Cu_2_Se film processed higher than the value obtained from other solutions (<0.1 mW m^−1^ K^−2^) is the highest value (≈0.5 mW m^−1^ K^−2^) reported among all flexible thermoelectric films so far [20]. Thin-film materials have one-dimensional nanoscale and two-dimensional macro-size, which is not only conducive to the microfabrication of macro-devices but also can take advantage of its one-dimensional nanosize effect. In addition, combined with ion beam, laser or photolithography technology, various artificially defined three-dimensional micro-nano structures can be produced, which is convenient for people to understand the relationship between thermoelectric properties and material structure in essence. Many methods have been reported for preparing Cu_2_Se thin films, such as electrochemical deposition [21,22,23], pulsed laser deposition [24,25], spin-coating [20,26,27], sputtering [28], thermal evaporation [29,30], etc. For the influence of these preparation methods on the film and the advantages of magnetron sputtering, please refer to our previous article [31]. Cu_2_Se films generally exhibit much lower thermoelectric properties, mainly due to structural defects, including voids and defects.

In this article, we report the preparation of Cu_2_Se thin films by pulsed magnetron sputtering on a glass substrate and systematically characterize the grown thin films using SEM, XRD, AFM, TEM, XPS and other means. We studied the thermoelectric properties of the film in the range of 300–400 K, and we chose this temperature range while considering the effect of the Cu_2_Se phase transition on its thermoelectric properties. The maximum Seebeck coefficient of Cu_2_Se film measured by ZEM-3 at 374 K is 33.51 µV K^−1^, and at 310 K, the maximum conductivity is 4.55 × 10^5^ S m^−1^. The conductivity of the film in this study is by far the largest of Cu_2_Se film. The maximum *PF* is 0.24 mW m^−1^ K^−2^ at 368 K. The film’s thermal conductivity is 2.14 W m^−1^ K^−1^ measured at room temperature using the TDTR method. The maximum *ZT* value is 0.073 at 374 K. With the rapid rise of miniature sensors and flexible electronic devices; flexible thermoelectric units may provide attractive solutions for power supplies. Cu_2_Se thin-film materials provide exciting advancements in flexible thermoelectrics for next-generation flexible electronics and sensors.

## 2. Experimental Section

### 2.1. Chemicals

Copper (I) selenide (Cu_2_Se, ~99.99%, specification: ϕ 50 mm × 4 mm, CAS: 20405-64-5) was purchased from ZhongNuo Advanced Material (Beijing, China) Technology Co., Ltd. Glass (BF33, CAS: 1344-09-8) base material and Si (n (100), 360 µm, CAS: 7440-21-3) base material was purchased from a company in Tianjin, China and the substrate is round, 4 inches in diameter.

### 2.2. Fabrication of Cu_2_Se Film

The equipment used in this experiment is the JCP-350 three-target magnetron sputtering-coating machine produced by Beijing Techno Technology Co., Ltd., in Beijing, China. The equipment is composed of the vacuum system, coating chamber, magnetron sputtering cathode, the substrate table, gas supply system, water cooling system, control and other components, and is mainly used for depositing a metal film, dielectric film and semiconductor film. Cu_2_Se is used to be the target material, which is a circular plane target, and the base material is glass and silicon. The surface magnetic induction intensity is about 300~400 Gs. The target power supply adopts 40 KHZ, 1 KVA asymmetric bipolar pulse power supply, and 500 VA radio frequency power supply, which can prepare various films. If the magnetic field is too small, it is not easy to produce glow discharge, and if the magnetic field is too large, it will affect the utilization rate of the target material and the uniformity of the produced film. The substrate table has the function of rotation and heating, which helps to improve the uniformity and adhesion of the film. In a typical preparation process, The positive pulse power supply of the bipolar pulse power supply is 80 V, the negative pulse power supply is 400 V, and the working current is 0.055 A. The vacuum degree in the sputtering chamber is 1.8 × 10^−3^ Pa before Ar gas is introduced, the vacuum degree in the chamber is 0.25 Pa during sputtering, the sputtering time is 10 min, and the substrate temperature is 333 K, 347 K, 413 K and 573 K four different temperature growth film. Substrate biasing is grounded, and the metal pressure pin presses the substrate to ensure that the insulating glass substrate and the silicon substrate are equipotential. The substrate is cleaned with H_2_SO_4_ and H_2_O_2_ (1:3) for 10 min at 100 °C.

### 2.3. Characterization of Cu_2_Se Film

We used X-ray diffraction (XRD, Bruker D8 ADVANCE, German Bruker Group (Karlsruhe, Germany). Cu target wavelength 1.5406 angstroms, tube current 40 mA, tube voltage 40 kV;) to explore the crystal structure of the synthesized Cu_2_Se. Use FIB (FEI Helios G4 CX, JEOL, in Tokyo, Japan) scanning to observe the surface morphology of the prepared film and measure the film thickness after FIB cutting. An atomic force microscope (AFM, Bruker Dimension Edge, German Bruker Group in Karlsruhe, Germany) was used to measure the roughness of the composite film. We used X-ray photoelectron spectroscopy to measure film surface elements and element valence states (American Thermoelectric Thermo Escalab 250 XI (Thermo Fisher Scientific, Waltham, MA, USA),related parameters: monochromatic Al Kα (hv = 1486.6 eV), power 150 W, 650 µm beam spot, voltage 14.8 KV, current 1.6 A, the charge calibration uses pollution carbon C 1s = 284.8 eV for calibration. We used a constant analyzer to measure pass energy (Ep), narrow sweep 20 eV, wide sweep 100 eV, vacuum degree 1 × 10^−10^ mbar). A JEOL JEM-2100 transmission electron microscope, Thermo Fisher Scientific, in Waltham, America was used to obtain TEM and electron diffraction images at an accelerating voltage of 200 kV. We used EDS energy spectrum scanning to measure the type and content of elements in the film.

### 2.4. Thermoelectric Performance Measurement of Cu_2_Se Film

The Seebeck coefficient and conductivity of the film were measured by ZEM-3 (ADVANCE RIKO, Inc. in Yokohama, Japan). We used time-domain thermal reflectance (TDTR) technology to measure the thermal conductivity of nanofilms. Time-domain thermal reflectance is a pump optics technique that can be applied to measure the thermal properties of materials. Cahill et al. have anteriorly depicted the implementation method of this technology and its application in research of the thermal conductivity of the film, the thermal conductivity of the interface, the spatially resolved measurement of microstructures, and the high-resolution mapping of the thermal conductivity of the diffusion multiple [32]. The thermal conductivity was calculated using the formula $e=\sqrt{\kappa \;\rho \;C_{v}}$, where *e* is the endothermic coefficient, *κ* is the thermal conductivity, *ρ* is the density, and $C_{v}$ is the specific heat of constant volume.

## 3. Results and Discussion

After preparing the Cu_2_Se thin film, the surface micro-morphology was observed with SEM, as shown in Figure 1a–d. Figure 1a is the result of observation under 600 times magnification. There were many particles on the surface of the film. The particles were almost uniformly distributed; the size of the particles was about 3–5 μm. Figure 1b is the surface of the film under 20,000 magnification. At this time, smaller particles could be observed on the film surface. The white particles shown in Figure 1c show a hexagonal structure with a particle size of 60–80 nm. A large number of random gray particles were slightly smaller in size than hexagonal particles. Figure 1d is a false cross-sectional SEM analysis of the film on Si substrate. It was found that the Cu_2_Se surface of magnetron sputtering in this experiment was relatively rough, and the thickness of the film was about 100 nm. To further understand the roughness of the film surface, an AFM observation was carried out, as shown in Figure 2a. The observed surface most fluctuation of the film in the range of 5 μm × 5 μm was about 92 nm, and the surface flatness of the film in most areas was about 20 nm.

To understand the crystal structure information of the Cu_2_Se thin film, a small-angle sweep scan was used. The 2θ range of the scan was 10–70 degrees. Figure 2 is the X-ray diffraction pattern of the Cu_2_Se thin-film sample (radiation = Cu Ka1, λ = 1.5406, Filter = Ni). Compared with the PDF standard card (PDF # 37-1187), the prepared Cu_2_Se film corresponds to the orthorhombic system. The thin-film sample was on the glass substrate, so there was a strong peak near the diffraction angle of 20 degrees. The diffraction peaks in the figure were consistent with the standard card, and abnormal peaks appear near the diffraction angles of 46.5 degrees and 51.814 degrees. After single-peak search analysis, it may be the diffraction peak of SiO_2_ in the glass substrate. The Cu_2_Se grain size was calculated by using the Scherer formula [33], taking $K_{1}$ = 0.89, the average grain size was: 15.9 nm. (lattice constant a = 1.41 nm, b = 2.04 nm, c = 0.41 nm):(1)$$D=\frac{K_{1}\lambda }{\beta cos\theta }$$

Here, $K_{1}$ is a constant; *λ* is the X-ray wavelength; *β* is the half-width of the diffraction peak; *θ* is the diffraction angle. In the above formula, the value of the constant $K_{1}$ is related to the definition of *β*. When *β* was the half-width, $K_{1}$ was 0.89; when *β* was the integral width, $K_{1}$ was 1.0.

The high-resolution transmission electron microscope (TEM) image of the cross-section of the film is shown in Figure 3a. The corresponding fast-reading Fourier transform (FFT) is shown in Figure 3b. 1, 2, and 3 in Figure 3b represent different crystal planes. The corresponding crystal plane spacing and crystal plane angles are shown in Table 1. Due to the unclear spots of selected area electron diffraction (SED), there was a little error in the angle and interplanar spacing. The good news was that the (211) and (060) crystal planes analyzed by TEM also appear in XRD analysis, indicating the rationality of XRD analysis and TEM analysis in this article. Figure 3c is the inverse Fourier transform of the electron diffraction pattern. It was found that there were crystal plane fractures on the dotted line, indicating the existence of dislocations in the thin-film crystal. Figure 3d–f is a high-angle circular dark-field image with an EDS spectrum scan of the cross-section of the film. The scan results showed that the copper–selenium ratio was 1.98:1.

The crystal structure information of the film was obtained through XRD. In order to know the element valence state of the film surface material, XPS analysis was performed. Figure 4 is the result of the XPS analysis. Figure 4a is a full-spectrum analysis. The full-spectrum analysis obtained the elements present in the material. The carbon element was the contaminated carbon added for charging calibration. In addition to the expected copper and selenium elements, there were also oxygen elements, indicating that the film’s surface may have been oxidized. Figure 4b shows the results of the sub-peak fitting of the high-resolution XPS spectrum of Cu 2p. The strong peaks fitted at 932.26 eV and 952.16 eV were attributed to Cu 2p 3/2 and 2p 1/2, which is consistent with the literature [34,35]. This indicates that the Cu ions in the Cu_2_Se thin film by magnetron sputtering still had the main valence state +1. The fitted weak peaks at 934.53 eV and 954.52 eV were attributed to Cu (ΙΙ), indicating that Cu (Ι) on the surface of the magnetron sputtered film was oxidized to Cu (ΙΙ). The weak satellite peaks with binding energies near 943.81 eV and 962.62 eV further confirmed the presence of a small amount of Cu (ΙΙ) on the surface of the film sample. Therefore, it can be seen that the oxygen element contained in the film may have been in the form of CuO. Figure 4c compares the peak area and peak intensity of the XPS spectrum after the fitting of Cu (Ι) and Cu (ΙΙ). After multiple fittings, the ratio of the content of Cu (Ι) and Cu (ΙΙ) was approximately 8:1 to 10:1. The high-resolution XPS spectrum of Se is shown in Figure 4d. After peak fitting, it was found that the 3d peak of Se contained two peaks Se 3d 5/2 and Se 3d 3/2, and the binding energy was 54.07 eV and 54.95 eV, respectively. Looking at the literature, it is clear that Se was not oxidized but existed in −2 valence. The element sensitivity factor analysis shows that Cu:Se on the surface of the film was 2.26:1, indicating that the surface film was missing Se. The element sensitivity analysis is shown in Formula (2), where n_1_ and n_2_ were element densities, *I*_1_ and *I*_2_ were the peak intensities of element characteristic peaks, and *S*_1_ and *S*_2_ were element sensitivity factors. The detailed Cu_2_Se film XPS peak fitting data are shown in Table 2.
(2)n1n2=I1S1I2S2

In this paper, the Seebeck coefficient/resistance measurement system introduced by Advance Riko, Japan, was used to evaluate the thermoelectric properties of Cu_2_Se thin films. As a feature of ZEM, both the Seebeck coefficient and resistance could be measured with one instrument. Figure 5 shows the temperature-dependent thermoelectric properties of thin-film samples grown at different substrate temperatures. The measurement temperature was between 300 and 400 K to study the influence of the phase change of Cu_2_Se film on its thermoelectric properties (350 K was the phase transition temperature of Cu_2_Se). As shown in Figure 5a,b, as the temperature increased, the conductivity of all films decreases, while the Seebeck coefficient increases. At a temperature of 300–400 K, the Seebeck coefficient and conductivity of the film at different substrate temperatures had little difference, and further rules need to be studied in a wider temperature range. The Seebeck coefficient and conductivity of the 300–400 K film were measured in great detail. Each temperature point was separated by 5 K. According to a previous study [19], it was seen that the abnormal Seebeck coefficient and conductivity of the bulk material near 350 K due to phase change. This film sample did not produce such a change, and more explanations need further research. The thermoelectric properties of films grown at different substrate temperatures had little difference, and the calculated power factor was 0.1–0.25 mW m^−1^ K^−2^.

This article is the first study on single target pulsed magnetron sputtering Cu_2_Se thin films. Compared with previous studies, the film’s conductivity in this article is 4.55 × 10^5^ S m^−1^, which is the largest among Cu_2_Se films measured so far. It may have been due to the higher carrier concentration and mobility of the thin-film sample, resulting in higher conductivity. Moreover, the measurement of the thermoelectric properties of thin-film samples has always been a challenge. During the measurement, it was found that the Seebeck coefficient and conductivity changed slightly under the same sample, the same measurement temperature, and the same temperature gradient. The Seebeck coefficient and conductivity measurement was based on the value obtained after reaching thermal equilibrium under a given temperature gradient. For the same sample, at the same temperature gradient, and at the same measurement temperature, the actual temperature gradient of the sample was different during each measurement, resulting in a difference in the measurement result. Therefore, these changes were ineffective and can be ignored. Under the same temperature gradient, the actual temperature difference between the two ends of the film after the film reached thermal equilibrium was related to the film properties. In addition, 21 temperature points were measured in the temperature range of 300–400 K in this article, which was more refined than all previous measurements on the thermoelectric properties of thin films. In the process of measuring the thermal conductivity of thin films using TDTR, the femtosecond pulsed laser was divided into pump and probe beams. Due to the pump laser pulse (λ = 400 nm), an instantaneous temperature rise was generated on the sample surface, and the temperature decay over time was monitored by the probe beam (λ = 800 nm). This transient decay curve was then fitted with a multilayer thermal model to obtain thermal conductivity. The thermal conductivity of films grown at different substrate temperatures was around 2 W m^−1^ K^−1^. This was higher than the thermal conductivity measured in the literature (Table 3). In a word, the measured thermal conductivity of Cu_2_Se film was low compare to other thermoelectric materials, which illustrated the potential of Cu_2_Se material as a thermoelectric material. Concerning the measurement of film thermal conductivity, the research group is still studying further. According to the measured thermal conductivity combined with the Seebeck coefficient and electrical conductivity, the calculated *ZT* value of the film is shown in Figure 6b. The *ZT* value of the film was small, mainly due to the low Seebeck coefficient of the film and the larger thermal conductivity than the value in the literature. Seebeck coefficient, electrical conductivity and thermal conductivity, etc., measured in this experiment were combined with the thermoelectric properties of the film in recent years, and the results are summarized, as shown in Table 3 below.

## 4. Conclusions

In summary, we prepared Cu_2_Se nanofilm by magnetron sputtering and systematically analyzed the physical and chemical properties of the film, such as the micromorphology, structure and composition. The surface of the material had the phenomenon of Se evaporation and Cu oxidation, and the ratio of copper to selenium was 2.26:1. The surface of the film observed on the macroscopic size had larger particles, which may have been caused by low vacuum (10^−3^ Pa). AFM measurement found that the surface of the film was rough, indicating that the prepared nanofilm introduced nanostructures with different length scales (length, roughness, etc.), which may effectively scatter in the entire phonon spectrum, thereby reducing the thermal conductivity of the material. In this study, the thermal conductivity of the film grown at different substrate temperatures at room temperature was measured at about 2 W m^−1^ K^−1^, which was equivalent to the thermal conductivity measured by Byeon et al. [19]. It is worth noting that although the thermoelectric properties of the film grown by changing the substrate temperature do not have a significant law, the electrical conductivity of the film was by far the largest. We believe that by increasing the absolute temperature, a higher *ZT* value could be obtained. This paper studies the possible influence of the phase change of the film on the thermoelectric performance in the range of 300–400 K. However, the measurement results in this experiment did not find an abnormal change in the Seebeck coefficient and conductivity due to phase change. The possible reasons need more experiments to find out. Our work fills the blank of Cu_2_Se nano-thermoelectric film and lays the foundation for further research on film thermoelectricity.

## Figures and Tables

**Figure 1 materials-14-02075-f001:**
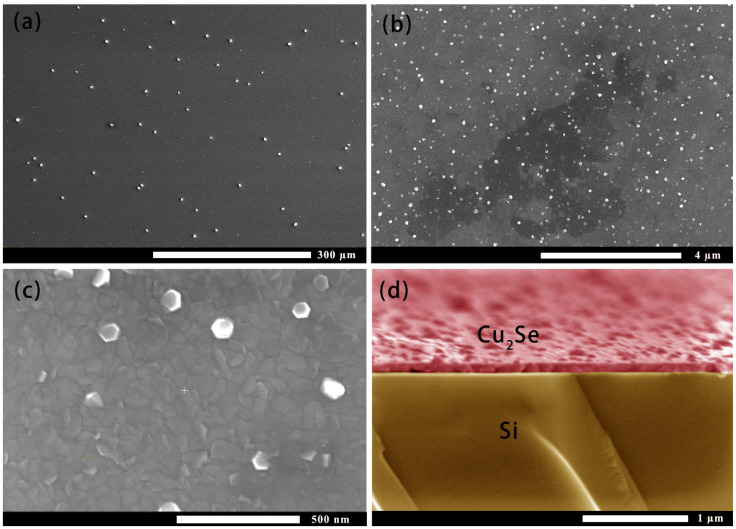
Scanning electron microscopic (SEM) image of the surface of the film. (**a**–**c**) Surface topography at different magnifications. (**d**) False cross-sectional SEM analysis of the film on Si substrate.

**Figure 2 materials-14-02075-f002:**
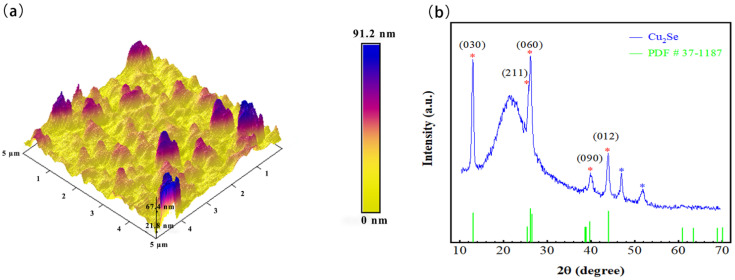
(**a**) Atomic force microscopy (AFM) images of Cu_2_Se. (**b**) X-ray diffraction (XRD) patterns of Cu_2_Se.

**Figure 3 materials-14-02075-f003:**
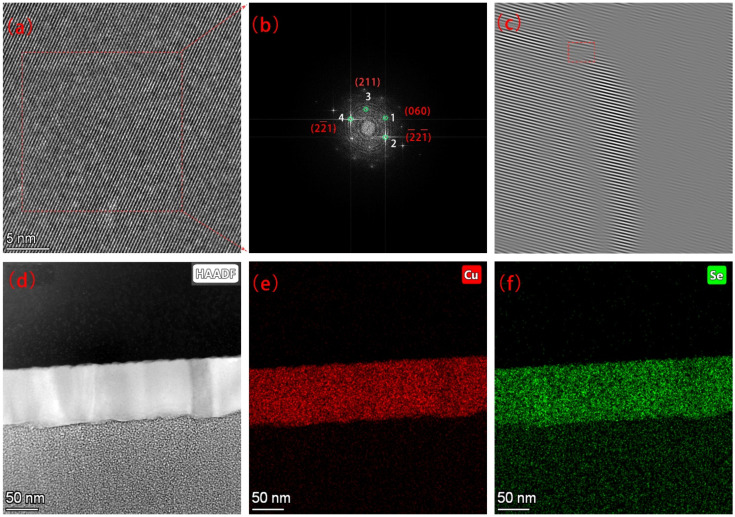
Structural characterization of the Cu_2_Se thin films. (**a**) The high-resolution transmission electron microscopic (TEM) image of the cross-sectional view of the thin film. (**b**) Fast Fourier-transform (FFT) of the TEM image in (**a**). (**c**) Inverse Fourier-transform of (060) crystal plane. (**d**) High-angle annular dark-field (HAADF) imaging technique in scanning TEM (STEM). (**e**,**f**) X-ray energy spectrum (EDS) analysis maps of Cu and Se taken from the Cu_2_Se.

**Figure 4 materials-14-02075-f004:**
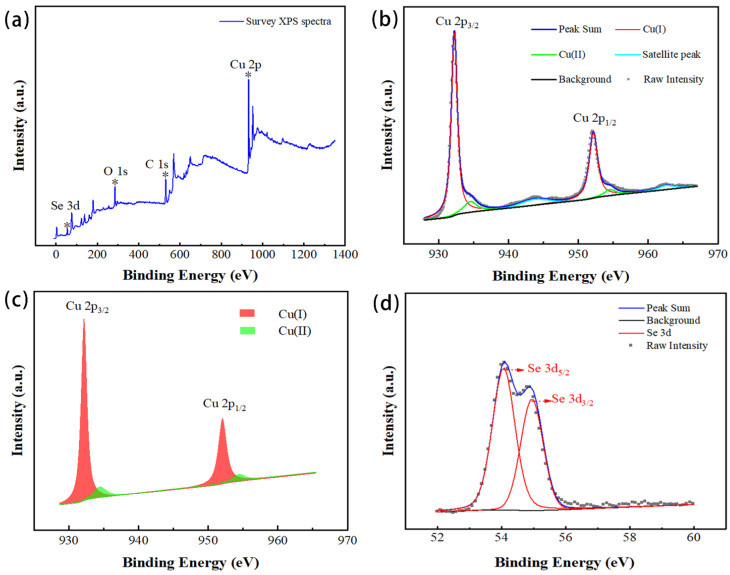
X-ray photoelectron spectroscopy (XPS) analysis of the Cu_2_Se thin film. (**a**) Survey. (**b**) Cu 2p. (**c**) Cu(Ι):Cu(ΙΙ). (**d**) Se 3d.

**Figure 5 materials-14-02075-f005:**
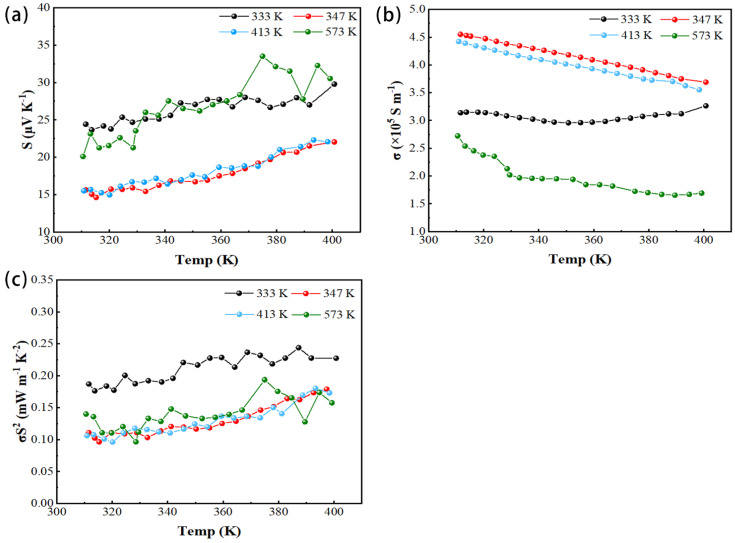
Thermoelectric properties of Cu_2_Se film on glass substrate prepared at various temperatures. (**a**) Seebeck coefficient *S*, (**b**) electrical conductivity *σ*, and (**c**) power factor (*PF*) *σS*^2^ of the film measured in the helium atmosphere.

**Figure 6 materials-14-02075-f006:**
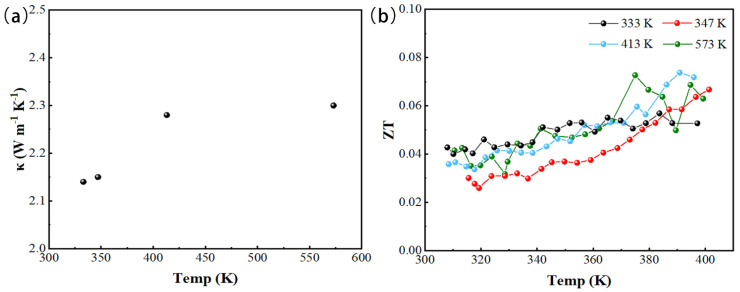
(**a**) Thermal conductivity values measured in multiple samples prepared at different temperatures, (**b**) The calculated ZT for the Cu_2_Se thin film.

**Table 1 materials-14-02075-t001:** d-Spacing values for the 4 local diffraction spots corresponding to the image.

Spot #	d-Spacing (nm)	Rec. Pos. (1/nm)	Degrees to Spot 1
**1**	0.3447	2.901	0.00
**2**	0.3338	2.996	57.75
**3**	0.3522	2.839	62.48
**4**	0.3338	2.996	122.25

**Table 2 materials-14-02075-t002:** Position BE (binding energy), the full width at half-maximum (FWHM) and area of Cu 2p, Se 3d and C 1s.

Peak Type	Position BE	FWHM	Area	Sens. Factor
Cu 2p3/2	932.26	1.02	244,279.7	26.513
Cu 2p1/2	952.16	1.43	122,139.8	
Se 3d5/2	54.07	0.83	6506.026	1.6
Se 3d3/2	54.95	0.82	4666.011	
C 1s	284.82	1.24	26,256.06	

**Table 3 materials-14-02075-t003:** Research progress of Cu2Se thin-film thermoelectric properties in recent years.

Methods	*S*	*σ*	*κ*	*PF*	*ZT*	Ref.
Chemical deposition		18				[36]
		17.8				[37]
Pulsed laser deposition	57	130		0.619		[25]
	60	10		0.35		[24]
Electrochemical deposition	80	27	0.77	0.173	0.07	[21]
Sputtering deposition	100	100	0.8	0.11	0.4	[28]
	34	450	2.12	0.25	0.073	This work
Spin-coating process	200–250	25	0.62	0.653	0.34	[26]
	80	100	0.4–1.4		0.14	[27]
Mechanical pressing	14.3	557.82	0.79	0.118	0.04	[17]
Wet-chemical process	50.8	104.7	0.25–0.3	0.27	0.3	[38]

Remarks: *S*: µV K^−1^, *σ*: ×10^3^ S m^−1^, *κ*: W m^−1^ K^−1^, *PF*: μW m^−1^ K^−1^.

## Data Availability

The data presented in this study are available on request. The presented data are not publicly available.

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
