# Peer review of "Thermoelectric Properties of Cu2Se Nano-Thin Film by Magnetron Sputtering"

_materials, 2021, doi:10.3390/ma14082075_

Round 1
Reviewer 1 Report
In this manuscript, the authors prepared and characterized Cu2Se thin films by magnetron sputtering relatively to its properties of thermoelectric performance. For that Seebeck coefficient, electrical conductivity, and thermal conductivity were also characterized.The work is clear and well-structured, and I can recommend this work for publication after the following revision:
- Please avoid use of capital letters in the middle of the sentence. See for example, in page 2, “…methods, such as simple drop casting (DC), vacuum filtration (VF) And vacuum filtration,…”, “And” should be changed to “and”. Change capital letter also in page 10, “…, Where e is the endo-thermic coefficient”. Please verify similar cases in whole manuscript.
- In page 3, do not repeat the words “target material” in the sentence:“The target material used is a Cu2Se target material”.
- In page 4, the authors wrote: “…and the thermal conductivity of the film is measured by TDTR. “Please define TDTR and take care that in page 10, wrote “This article uses Time Domain Thermal Reflectance (TDTR) technology to measure the thermal conductivity of nano-films.”. This last sentence is not necessary since this information should already be very detailed in the Materials and Methods section.
- In page 9, please delete the extra “.” in the middle of the sentence: “…indicating that the abnormal Seebeck coefficient and conductance measured by Byeon et al[19]. cannot be measured under a single temperature gradient. rate.”
- In page 6, the authors claim:”... and the corresponding fast-reading Fourier trans-form (FFT) (Figure 3 (b)) shows multiple sets of diffraction points, indicating that the magnetron sputtering The shot film is polycrystalline.” Please explain better this sentence.
- In page 10, the authors claim: “During the measurement, it was found that the Seebeck coefficient and conductivity will change slightly under the same sample, the same measurement temperature, and the same temperature gradient.” A possible explanation for such evidence must be added to the manuscript. Are these changes effective? Please explain.
- In page 10, please rephrase the sentence” In recent studies on the thermoelectric properties of Cu2Se thin films, the preparation methods of the films are different, such as electrochemical deposition, pulsed laser deposition, spin coating, thermal evaporation, etc.”
- In page 10, a table indicating the achieved values and the ones already in literature will be welcome.
- Rephrase this sentence: “This article uses Time Domain Thermal Reflectance (TDTR) technology to measure the thermal conductivity of nano-films.”
- Please indicate the error bars of the values presented in figures 5 and 6. Please explain better why the Seebeck coefficient increase with the temperature and then go back to the values obtained at the lowest temperature (Figure 5a)).
- The manuscript must be read carefully and the English improved.
Reviewer 2 Report
The authors present thermoelectric properties of Cu2Se nano-film by magnetron sputtering. The authors deposited 90-130 nm Cu2Se films via magnetron sputtering and characterized the films via SEM, XRD, AFM, TEM and XPS. Thermoelectric properties were studies by probing the Sebeck effect and the ZT values of the deposited films. The manuscript has potential; however, there are some technical and graphical errors in the manuscript that need serious attention. Hence, I would like decide based on the major revision of the following comments.
1- There are serious grammatical and structural errors in the manuscript. Check the manuscript with a native English speaker or a Linguistic expert for clarity.
2- There is a sentence in the abstract which says, "XRD and TEM analysis show that the film is low-temperature α-Cu2Se, ". what is the intended meaning?
3- Briefly explain the pros and cons of different methods of Cu2Se thin film deposition and explain the reason why the authors preferred magnetron sputtering over the rest of the deposition methods.
4- Provide the CAS numbers of all the chemicals and substrates in the experimental section.
5- Provide the base book bibliographic reference of Scherrer formula in eq 1.
6- The grain size of the film in Figure 1(c) and Figure 3(d) look different. Explain the reason for this discrepancy.
7- Provide a comparative analysis of thermoelectric properties and ZT values of the present work with the recently published papers in a tabular form.
Reviewer 3 Report
This paper has interest because it is new in applying pulsed magnetron sputtering of Cu2Se to theroelectric films. However, it needs significant improvement before it can be published.
see attached file

Round 2
Reviewer 1 Report
This manuscript has been improved considering the reviewers' suggestions and, therefore, I can recommend this work for publication.
Reviewer 2 Report
The authors have provided the revised version of the manuscript. The paper looks in good shape after the revision. I would like to accept paper publication after the minor revision of the following comments.
1- There are still many grammatical errors in the paper. Moderate English editing still required for perfection.
2- Provide the CAS numbers for Glass and Si as well.
3- Always provide the line number and page number of the changes made in the manuscript for reviewers' and editor's convenience.
Good Luck
Reviewer 3 Report
Please see attached file
